# The RSL3 Induction of *KLK* Lung Adenocarcinoma Cell Ferroptosis by Inhibition of USP11 Activity and the NRF2-GSH Axis

**DOI:** 10.3390/cancers14215233

**Published:** 2022-10-25

**Authors:** Wenlong Zhang, Xiaohe Li, Jiaqian Xu, Ying Wang, Zhengcao Xing, Shuming Hu, Qiuju Fan, Shaoyong Lu, Jinke Cheng, Jianmin Gu, Rong Cai

**Affiliations:** 1Department of Biochemistry & Molecular Cell Biology, Key Laboratory of Cell Differentiation and Apoptosis of Chinese Ministry of Education, School of Medicine, Shanghai Jiao Tong University, Shanghai 200025, China; 2Shanghai Institute of Immunology, School of Medicine, Shanghai Jiao Tong University, Shanghai 200025, China; 3Department of Pathophysiology, Key Laboratory of Cell Differentiation and Apoptosis of Chinese Ministry of Education, School of Medicine, Shanghai Jiao Tong University, Shanghai 200025, China; 4Department of Biochemistry & Molecular Cell Biology, State Key Laboratory of Oncogenes and Related Genes, Renji Hospital Affiliated, Shanghai Key Laboratory for Tumor Microenvironment and Inflammation, Shanghai Jiao Tong University School of Medicine, Shanghai 200025, China; 5Department of Thoracic Surgery, Zhongshan Hospital, Fudan University, Shanghai 200025, China

**Keywords:** NRF2, RSL3, USP11, *KLK* LUAD, ferroptosis

## Abstract

**Simple Summary:**

High NRF2 level confers *KLK* LUAD cell resistance to ferroptosis. Here, we showed that the inhibition of NRF2-GSH axis sensitized a small molecule RSL3 to induce *KLK* LUAD cell ferroptosis in vitro. RSL3 treatment inhibited activity of the NRF2-GSH signaling during *KLK* LUAD cell ferroptosis in vitro and in vivo. The mechanism is that RSL3 was able to directly bind to USP11, a recently identified de-ubiquitinase of NRF2, and inactivate USP11 protein to induce NRF2 protein ubiquitination and degradation in *KLK* LUAD cells. It was discovered for the first time that RSL3 induction in *KLK* LUAD cell ferroptosis by the suppression of USP11-NRF2-GSH signaling, in parallel to GPX4 inhibition.

**Abstract:**

Antioxidant transcription factor NRF2 plays a pivotal role in cell ferroptosis. *KLK* lung adenocarcinoma (LUAD) is a specific molecular subtype of *Kras*-mutant LUAD. The activation of mutant *Kras* in combination with the inactivation of *Lkb1* and *Keap1* abnormally increases NRF2 expression, while high NRF2 confers *KLK* LUAD cell resistance to ferroptosis. This study assessed the inhibition of NRF2-GSH axis to sensitize a small molecule RSL3 to induce *KLK* LUAD cell ferroptosis and then explored the underlying molecular mechanisms. The data showed that the NRF2-GSH inhibition sensitized RSL3 induction of *KLK* LUAD cell ferroptosis in vitro, while RSL3 treatment reduced level of NRF2 protein in *KLK* LUAD during ferroptosis. Moreover, RSL3 treatment inhibited activity of the NRF2-GSH signaling during in *KLK* LUAD cell ferroptosis in vitro and in vivo. Mechanistically, the RSL3 reduction of NRF2 expression was through the promotion of NRF2 ubiquitination in *KLK* LUAD cells. In addition, RSL3 was able to directly bind to USP11, a recently identified de-ubiquitinase of NRF2, and inactivate USP11 protein to induce NRF2 protein ubiquitination and degradation in *KLK* LUAD cells. These data revealed a novel mechanism of RSL3 induction in *KLK* LUAD cell ferroptosis by suppression of the USP11-NRF2-GSH signaling. Future study will confirm RSL3 as a novel therapeutic approach in control of *KLK* lung adenocarcinoma.

## 1. Introduction

A single amino acid substitution in the KRAS protein due to a single nucleotide substitution mutation activates this pro-oncogene and transforms normal cells to tumor cells, including lung adenocarcinoma (LUAD). *Kras*-mutant lung adenocarcinoma accounts for about 30% of LUAD, which has poor prognosis and urgently needs effective treatment methods in the clinic [1]. Clinically, *Kras*-mutant LUAD is dominated in three robust subsets of molecular lung cancer classifications, among which inactivation of the liver kinase B1 [*Lkb1*; or named as Serine/threonine kinase 11 (*Stk11*)] is the most frequently co-occurring events [2,3,4]. *Lkb1* loss induces bronchial cell energetic and redox stress and in turn, inactivates *Keap1* but stabilizes NRF2 (NFE2L2, nuclear factor erythroid 2-related factor 2) in LUAD cells (*KLK* LUAD) [5]. Meanwhile, *Kras* mutation is reported to augment *Nrf2* mRNA levels in tumor cells [6] and subsequently, to induce strong anti-oxidation and metabolic adaptation in *KLK* LUAD cells in favor of their survival [5].

Ferroptosis is an iron-dependent regulated cell death (RCD), defined by Stockwell Lab in 2012 [7], while NRF2 plays a key role in iron metabolism and cell redox homeostasis by transcriptionally regulating the expression of the related downstream genes [8,9] due to its regulation of many ferroptotic cascade components, defending against oxidative stress and alleviating lipid peroxidation during ferroptosis [10]. Alterations of NRF2 contribute to tumorigenesis and tumor metastasis and progression [9], first reporting a key role in protecting hepatocellular carcinoma (HCC) from ferroptosis [11]. Research showed that in lung cancer, the proteasomal degradation of NRF2 protein promoted by the E3 ligase MIB1 could sensitize lung cancer cells to ferroptosis [12] and that the suppression of NRF2 expression and activity could be a novel approach to controlling LUAD clinically. In addition, RSL3, a small molecule, induces ferroptosis through synthetic lethal interactions with mutant *Ras* gene in tumor cells [13]. RSL3 was able to inhibit activity of glutathione (GSH, r-glutamyl cysteingl + glycine) peroxidase 4 (GPX4) at the active site of selenocysteine [14], leading to cell ferroptosis. RSL3 possesses a chloroacetamide moiety, essential for its activity, that can target enzymes with a nucleophilic active site, like serine, threonine, cysteine or selenocysteine [14]. As we know, the GSH-GPX4 axis was able to catalyze the lipid-OOH to the lipid-OH and GPX4 depletion or inactivation will sensitize cells to ferroptosis [14,15]. Using the active RSL3 affinity probe, it was reported that RSL3 could directly bind to GPX4 to inactivate its peroxidase activity, which in turn sensitizes breast cancer cells to ferroptosis [14]. 

Again, ubiquitination is the chief mechanism in regulating NRF2 stability and expression, and three E3 ubiquitin ligase complexes were proved as the main controllers of the ubiquitination and proteasomal degradation of NRF2 [16,17,18]. Ubquitination is a dynamic and reversible process and NRF2 de-ubiquination was recently reported through the ubiquitin-specific-processing protease 11 (USP11) execution in non-small cell lung cancer cells [19]. The de-ubiquitinating enzymes (DUBs) are a class of the enzyme family of cysteine proteases and metalloproteases [20]. The former has an active site with cysteine, which might be the potential targets of RSL3. 

In this study, we described that by restricting USP11 activity, RSL3 inhibited NRF2-GSH axis, in parallel to GPX4 inhibition, to induce *KLK* LUAD cell ferroptosis. We expect to provide pivotal information regarding the clinical use of RSL3 in the treatment of *KLK* LUAD in the future. 

## 2. Materials and methods

### 2.1. Cell Lines and Culture

Human *KLK* LUAD cell lines A549 and H2122 were originally obtained from the American Type Culture Collection (ATCC; Manassas, VA, USA). A549 cells were maintained in F-12K medium (Gibco, Gaithersburg, MD, USA) supplemented with 10% fetal bovine serum (FBS; Gibco) and 1% penicillin/streptomycin (Sigma Chemicals, St. Louis, MO, USA), while H2122 cells were grown in Roswell Park Memorial Institute medium-1640 (RPMI-1640; Gibco) with 10% FBS (Gibco) and 1% penicillin/streptomycin (Sigma) in a humidified incubator with 5% CO_2_ at 37 °C. 

### 2.2. NRF2 siRNA, USP11 Vector, and Cell Infection to Establish Stable LUAD Cell Sublines

Lentivirus carrying NRF2 siRNA or negative control siRNA was obtained from Genomeditech company (Shanghai, China) and used to generate the NRF2-knocked down A549 and H2122 cell sublines from our previous study [21]. To establish stable NRF2 overexpression LUAD cell sublines, A549 and H2122 cells were infected with PCDH-NRF2 lentivirus and then selected in puromycin-containing medium for 1 week. In this study, we also amplified the coding sequences of human *Usp11* gene using PCR and subcloned the PCR products into a PGMLV-CMV-MCS-3*Flag-PGK-Puro vector. The empty vector was used as the negative control. After DNA sequence confirmation, these vectors were transfected into HEK293 cells, and packaged into lentiviruses. The lentiviruses were utilized to infect A549 and H2122 cells and selected the cells with the puromycin-containing media for one week to establish the USP11 overexpression (USP11-OE) A549 and H2122 cell sublines.

### 2.3. RNA Isolation and RNA-Seq

Total cellular RNA was isolated from A549 and shNRF2 A549 cells using the Trizol reagent (Invitrogen, Carlsbad, CA, USA) and submitted to KangChen Biotech (Shanghai, China) for RNA-Seq. The RNA-seq library was prepared using Illumina kits and then subjected to the Kyoto Encyclopedia of Genes and Genomes (KEGG) pathway and network analysis using the Illumina Hiseq 4000 [21].

### 2.4. Quantitative Reverse Transcriptase-Polymerase Chain Reaction (qRT-PCR)

TRIzol-isolated total cellular RNA was reversely transcribed into complementary DNA (cDNA) using the Fast King gDNA Dispelling RT SuperMix kit (see Table 1). Quantitative PCR was then performed using the Light-Cycler 480 (Roche, Basel, Switzerland) with the TB Green Premix (see Table 1). Level of the 18S rRNA was used as an internal control and mRNA levels were quantified using the 2-^ΔΔ^Ct method. The experiment was in triplicate and repeated at least once. The primers used for qPCR are HMOX1, 5′-AAGACTGCGTTCCTGCTCAAC-3′ and 5′-AAAGCCCTACAGCAACTGTCG-3′; GCLC, 5′-GGAGGAAACCAAGCGCCAT-3′ and 5′-CTTGACGGCGTGGTAGATGT-3′; GCLM, 5′-TGTCTTGGAATGCACTGTATCTC-3′ and 5′-CCCAGTAAGGCTGTAAATGCTC-3′; GPX4, 5′-GAGGCAAGACCGAAGTAAACTAC-3′ and 5′-CCGAACTGGTTACACGGGAA-3′; SLC7A11, 5′-TCTCCAAAGGAGGTTACCTGC-3′ and 5′-AGACTCCCCTCAGTAAAGTGAC-3′; FTH1, 5′-CCCCCATTTGTGTGACTTCAT-3′ and 5′-GCCCGAGGCTTAGCTTTCATT-3′; FTL, 5′-CAGCCTGGTCAATTTGTACCT-3′ and 5′-GCCAATTCGCGGAAGAAGTG-3′; PTGS2, 5′-CGGTGAAACTCTGGCTAGACAG-3′ and 5′-GCAAACCGTAGATGCTCAGGGA-3′; 18 s, 5′-GAAACGGCTACCACATCC-3′ and 5′-CACCAGACTTGCCCTCCA-3′.

### 2.5. Western Blot

Cells were washed in ice-cold phosphate-buffered saline (PBS) and lysed in the radioimmunoprecipitation assay buffer (RIPA buffer; 150 mM NaCI, 50 mM Tris base, 0.1% sodium dodecyl sulfate, 1% Triton-X-100 at pH7.4) containing the protease inhibitors and phenylmethane sulfonyl fluoride (PMSF) on ice for 20 min. The cell lysates were then ultrasonicated and centrifuged at 12,000 rpm for 15 min at 4 °C to collect the supernatants. Equal amounts of these protein samples were loaded on the sodium dodecyl sulfate–polyacrylamide gel electrophoresis (SDS-PAGE) gels and electronically separated and transferred onto polyvinylidene fluoride membranes (PVDF; Millipore, Billerica, MA, USA). For Western blotting, the membranes were first incubated in 5% bovine serum albumin/Tris-based saline (TBS) solution at room temperature for 1 h and then with the primary antibodies (Table 1) at 4 °C overnight. On the next day, the membranes were washed in TBS-Tween 20 (TBS-T) three times for 10 min each and further incubated with a horseradish peroxidase (HRP)-conjugated secondary antibody (Table 1) at the room temperature for 1 h. Protein bands were visualized using the Enhanced chemiluminescence (ECL) substrate and quantified using the Image-Quant LAS 4000 (GE Healthcare, Fairfield, CT, USA). Full Western blot images can be found at Appendix A. 

### 2.6. Cell Viability Assay

Changes in cell viability were assayed using the Cell Counting Kit-8 kit (CCK-8; Table 1) according to the manufacturer’s instructions. In brief, *KLK* LUAD cells were cultured and subjected to infection (see above) and then reseeded into 96-well plates for growth for up to 1 day. At the end of each experiment, cell culture was added with 20 μL of the CCK-8 reagent and the microplate was further incubated at 37 °C for 2 h. After that, cell optical density (OD) was measured at 450 nm using a microplate reader. The relative cell viability of each group was calculated against the control cells (n = 3).

### 2.7. Measurement of GSH (Glutathione) and GSSG (Glutathione Oxidized) Levels in Cells

The levels of GSH and GSSG were detected using a GSH and GSSG Assay Kit (Table 1) according to the manufacturer’s protocol. In particular, *KLK* LUAD cells were grown and subjected to infection (see above). Next, cells were washed in ice-cold PBS and collected after centrifuging and then added with the protein removal reagent M solution and put into a −20 °C freezer for the twice frozen-thaw in liquid nitrogen and the 37 °C water bath alternatively. After that, the cell samples were put into ice-bathing for 5 min and then centrifuged at 10,000× *g* at 4 °C for 10 min to collect the supernatants, which were used for assay of GSH, GSSG, and GSH+GSSG levels, i.e., the samples were added with 150 μL of the total glutathione detection solution and thoroughly mixed and incubated at the room temperature for 5 min. Next, the samples were added 50 μL of the 0.5 mg/mL NADPH solution and incubated at room temperature for 25 min for a spectrophotometer measurement of the OD number at 412 nm. The experiment was in triplicate and repeated at least once. 

### 2.8. Lipid Peroxidation Assay

Changes in cell lipid peroxidation level were assayed using the BODIPY™ 665/676 staining kit (Table 1). After infection, the cells were washed in ice-cold PBS and then added with 1 μM of the BODIPY™ 665/676 and then incubated at 37 °C for 30 min. After that, cells were washed in PBS thrice and analyzed by using the Flow cytometer (BD Biosciences, Franklin Lakes, NJ, USA).

### 2.9. Animal Experiment

The animal experiments of the study were approved by the Animal care Committee of School of Medicine, Shanghai Jiao Tong University (Shanghai, China) and all experiments were performed following the Guidelines of the Care and Use of Laboratory Animals issued by the Chinese Council on Animal Research. In brief, 8-week-old male nude mice from Charles River Laboratories (Shanghai, China) were maintained in a specific pathogen-free (SPF) “barrier” facility and housed under controlled temperature and humidity and alternating 12-h light and dark cycles. The mice will receive SPF mouse chow and be allowed to drink sterile water ad libitum. After two weeks in house, the mice were subcutaneously injected with A549 cells (100 μL containing 5 × 10^6^ cells/injection) and monitored for tumor cell xenografts to reach approximately 100 mm^3^. The mice were then divided into two groups (n = 5), the RSL3 treatment (100 mg/kg; dissolved in 5% dimethyl sulfoxide/corn oil; administrated intratumorally twice a day for one week) and control (5% dimethyl sulfoxide/corn oil only) groups. After that, the mice were sacrificed, and tumor weight was measured respectively.

### 2.10. Detection of Malondialdehyde (MDA) Level

Tissue level of MDA in tumor cell xenografts was detected using the MDA Assay Kit (Table 1) according to the manufacturer’s protocol. Tumor cell xenograft tissues were mechanically ground and lysed in the RIPA buffer containing proteases inhibitors and the insoluble tissue debris was removed after centrifuging. For detection of MDA level, the samples were mixed with thiobarbituric acid and incubated for 1 h at 95 °C and the the optical density (OD) number was measured at 532 nm.

### 2.11. Immunohistochemistry

Tumor cell xenografts were processed and embedded in paraffin routinely and prepared for 4-µm-thick tissue sections. For immunohistochemistry, the sections were and deparaffinized and rehydrated into tap water. After that, sections were blocked in 3% bovine serum albumin (BSA)/PBS at room temperature for 1 h and then incubated with primary antibodies (see Table 1) at 4 °C overnight. On the next day, the sections were washed thrice with PBS and incubated with an HRP-conjugated secondary antibody according to the manufacturer’s instructions for 1 h at room temperature and subsequently visualized with the 3,3′-diaminobenzidine (DAB) solution. 

### 2.12. Immunoprecipitation (IP)-Western Blot

Cells were lysed in the IP buffer (150 mM NaCI, 50 mM Tris-base, and 1% NP40 at pH7.4) supplemented with 1% protease inhibitors and PMSF on ice for 30 min. After that, cells were centrifuged at 12,000× *g* at 4 °C for 15 min to collect the supernatants. The latter was added with normal IgG plus 20 μL of protein A/G beads and mixture was incubated at room temperature for 1 h and centrifuged to 12,000× *g* at 4 °C for 15 min to collect the supernatants. The resulted supernatants were incubated with a primary antibody (1 μL) at rotation for 16 h at 4 °C and then added with 20 μL of protein A/G beads and incubated at rotation for 3 h at 4 °C. Next, the beads were then washed five times with the IP buffer through centrifuging. The immunoprecipitates were afterward treated with 100 μL of 2% SDS solution and analyzed using Western blot. 

### 2.13. Molecule-Docking Analysis

The crystal structure of the USP7 catalytic domain (PDB ID:5NGE) was selected as the template for the homology modeling of gene association analysis according to their sequence homology. The USP11 structure was predicted using the Swiss-MDOEL online software to construct the small molecule-protein complex. After that, we applied the Schrodinger software for the analysis of the small molecules with the corresponding protein. In this docking process, we first utilized the Preparing Protein function of Schrodinger software to optimize the protein structure for the next docking. After that, we applied the Ligand Preparation for the given small molecules to predict their binding affinity. In addition, we utilized the function of the gride generation in the software to generate the grid files for the docking analysis. Finally, the molecular docking analysis was carried out by using the glide docking function, while the small molecule screening was through the docking scores. In the results, the combination with a score close to −7 was regarded as the best combination of the small molecule-protein binding and we then selected the best structure for the initial structure of simulation data. 

### 2.14. Purification of Wild Type and Mutant USP11 Proteins

The MaxCodon^TM^ Optimization Program (Version 13; MERRYBIO, Nanjing, China) was utilized to optimize the amino acid sequences of USP11 protein. We constructed the coding sequences of the wild type and mutant *USP11* (Y833E, Q402E) using PCR and subcloned into the expression vector P35. The resulted vectors were confirmed using the restrict enzyme digestion and DNA sequencing and thereafter, transformed into human embryonic kidney 293 (HEK293) cells for the expression of USP11 and USP11-1 proteins, which were then purified by the affinity chromatography, i.e., the cell culture medium was obtained after 5 days of vector transfection and dissolved in 50 mM Tris, 150 mM NaCl at pH8.0, 8 M urea, 20 mM imidazole buffer and injected into the Ni-IDA column for purification. Subsequently, the protein solutions were added to a dialysis bag, and re-natured in a buffer solution containing 1 × PBS at pH7.4, 4 mM GSH, 0.4 mm GSSG, 0.4 mL arginine, and 1 M urea at 4 °C. Next, the USP11 and USP11-1 proteins were dialyzed in a storage solution (1 × PBS at pH7.4) for approximately 6 to 8 h. The supernatant was filtered by a 0.22 μm filter and stored at −80 °C for further analysis. 

### 2.15. Affinity Binding Assay

The Octet platform was used to detect and analyze molecule interactions based on bio-layer interferometry (BLI). The biotin-labeled USP11 and mutant USP11-1 proteins were conjugated with the super streptavidin (SSA) biosensors, while RSL3 was diluted in PBS)-Tween 20 (PBS-T) to 218 μM, 109 µM, 54.5 µM, 27.3 µM, 13.6 µM, and 6.81 µM of concentrations. After that, different RSL3 dosed solutions and the USP11 and USP11-1 proteins were added into a 96-well plate and after well mixed, the interaction was detected at 1000 rpm and 30 °C. The 1:1 fitting model was selected, i.e., one molecule USP11 or USP11-1 protein bound to one RSL3 molecule to be reached. The fitting method was globally analyzed for six different RSL3 concentrations as a group for their associations with USP11 or USP11-1 protein.

### 2.16. Statistical Analysis

All data were expressed as mean ± SD for statistical analysis, which was performed by using SPSS 17.0 (SPSS, Chicago, IL, USA). Student’s *t* test was conducted to compare the means between two different groups, while a *p* value equal to or less than 0.05 was considered statistically significant. 

## 3. Results

### 3.1. RSL3 Induction of KLK LUAD Cell Ferroptosis Sensitized by the NRF2-GSH Inhibition

In this study, we first performed RNA sequencing (RNA-seq) analysis, and our data showed that the ferroptosis pathway was the top hit for downregulation after the KEGG enrichment analysis in NRF2 knocked down the A549 cell line (Figure 1A) [21]. The downregulated gene expression in the ferroptosis pathway included *Slc7a11*, *Gclc* and *Gclm* after knockdown of NRF2 expression and restoration were verified (Figure 1B,C) [21]. Due to the GSH synthesis via the NRF2 target genes, we analyzed the reduced GSH and oxidized GSSG levels in A549 and H2122 cells after the knockdown of NRF2 expression and restoration. We found that the GSH level was lower after the knockdown of NRF2 expression, although the oxidized GSSG level was not significantly altered (Figure 1D). However, after NRF2 expression was restored in A549 and H2122 cells, the reduced GSH level was significantly recovered (Figure 1D), indicating that NRF2 was able to regulate GSH level in *KLK* LUAD cells.

We thus speculate that the NRF2-GSH axis could be essential for *KLK* LUAD cell ferroptosis. To prove this, we first treated A549 and H2122 cells with 8 μM RSL3 for 12 h and assayed cell death level using a phase contrast microscope and cell viability CCK8 assay, and then detected the Fe^2+^ concentration in cells using the FerrOrange, lipid hydroperoxide (LPO) level using the BODIPY 665/676, and mitochondria morphology changes using transmission electron microscope (TEM) (Appendix A). These experiments demonstrated that a high RSL3 dose could induce *KLK* LUAD cell ferroptosis. In addition, we treated *KL* LUAD cell line H23 and *K-ras* wild-type LUAD cell line H1299 with RSL3 (2–8 μM), and found that RSL3 induced death of both H23 and H1299 cells in a dose-dependent manner, which can be rescued by Deferoxamine mesylate (DFO; 100 μM) treatment. More importantly, we found that RSL3 was more efficient in H23 and H1299 cells than in A549 and H2122 cells, i.e., 8 μM of RSL3 induced approximately 75% and 60% H23 and H1299 cells to death individually, but only induced less than 50% A549 and H2122 cells to death (Appendix A).

We further found that 12 h RSL3 treatment (8 μM) in NRF2 knocked down *KLK* LUAD cells sensitized RSL3 for induction of tumor cell death, whereas restoration of NRF2 expression was able to rescue or reverse death of A549 and H2122 cells (Figure 1E). Moreover, we proved that DFO (100 μM), a selective ferroptosis inhibitor, could rescue the cell death at least partially (Figure 1F). In addition, we added 5 mM NAC, a precursor of cysteine to the indicated cells (Figure 1F), and observed that the RSL3-induced death difference in A549 and H2122 cells, caused by NRF2 expression was narrowed, indicating the effects of NRF2 were at least partially GSH-dependent.Our flow cytometric assay revealed that knockdown of NRF2 expression significantly increased cellular LPO level in A549 and H2122 cells after RSL3 treatment (Figure 1G), illustrating an importance of NRF2 protein in the regulation of *KLK* LUAD cell ferroptosis.

### 3.2. RSL3 Reduction of NRF2 Expression in KLK LUAD Cells during Tumor Cell Ferroptosis

In this study, we confirmed the RSL3 induction of *KLK* LUAD cell ferroptosis and also found RSL3 reduction of NRF2 protein expression in a dose dependent fashion (Figure 2). Moreover, levels of NRF2-downstream proteins, like SLC7A11 and GCLC, both of which participate in the GSH synthesis, were also reduced after RSL3 treatment in *KLK* LUAD cells (Figure 2). RSL3 might inhibit SLC7A11 expression through NRF2; thus, we measured the cystine uptake in A549 and H2122 cells before and after RSL3 treatment, and the results revealed that RSL3 could also significantly reduce the cystine uptake, as did Erastin (Appendix A). However, level of GPX4 protein that is able to directly bind to RSL3, was not significantly changed after RSL3 treatment of A549 and H2122 cells (Figure 2). Furthermore, more interestingly, as the control, we did not find FIN I Erastin inhibition of NRF2 protein expression in A549 and H2122 cells (Figure 2), indicating that further investigation is needed to assess the underlying mechanisms of RSL3 inhibition of NRF2 expression in *KLK* LUAD cells.

### 3.3. RSL3 Inhibition of the NRF2-GSH Axis during Ferroptosis in KLK LUAD Cells In Vitro and In Vivo

We further treated A549 and H2122 cells with RSL3 at different dosages (1 μM, 4 μM, and 8 μM) and found that RSL3 suppressed expression of NRF2 protein in a dose-dependent manner (Figure 3A) but did not affect levels of *Nrf2* mRNA (Figure 3B). Moreover, a low RSL3 dose (1 μM) even increased levels of NRF2 protein, but a high RSL3 dose (8 μM) was able to significantly reduce NRF2 protein in *KLK* LUAD cells (Figure 3A). As we know, the NRF2-targeting genes are involved in ferroptosis pathway, RSL3 treatment also reduced their levels in *KLK* LUAD cells in comparison with DMSO treatment, except *Hmox1* (Figure 3C). Meanwhile, the GSH level was also decreased in A549 and H2122 cells after RSL3 treatment (Figure 3D), indicating that RSL3 inhibited NRF2 and GSH during RSL3 induced *KLK* LUAD cell ferroptosis. In order to prove that the RSL3-induced reduction of GSH was dependent on NRF2 expression, we established stable NRF2 overexpression in A549 and H2122 cells (Figure 3E) and demonstrated that NRF2 overexpression could significantly alleviate the GSH level as well as GSH/GSSG ratio reduction caused by RSL3 treatment (Figure 3F).

To confirm these in vitro data in vivo, we injected A549 cells into immunodeficient nude mice and treated them with RSL3 for 1 week after the tumor cell xenograft volume reached 100 mm^3^. The data in Figure 4A showed that RSL3 significantly restricted growth and weight of tumor cell xenografts in nude mice. To confirm tumor growth constrained by RSL3 treatment in vivo for induction of tumor cell ferroptosis, we assessed the MDA (malondialdehyde) level in tumor cell xenografts and the former is the marker of the lipid peroxidation in cells and represents the ferroptosis content. We found that RSL3 treatment upregulated MDA level in tumor cell xenografts (Figure 4B). RSL3-treated mice also had an obvious lower level of NRF2 protein than DMSO treated group in tumor cell xenografts (Figure 4C,D), shown by both Western blot and immunohistochemistry results, which was also true for NRF2-targeting genes in vivo (Figure 4E).

### 3.4. RSL3 Reduction of NRF2 Expression by Promoting Its Ubiquination in KLK LUAD Cells

Our data revealed that RSL3 reduced NRF2 expression at the protein level but not at the mRNA level (Figure 3A,B), indicating that changed protein ubiquination activity may be involved. Thus, we treated A549 and H2122 cells with a proteasome inhibitor MG132 in addition to RSL3 treatment. The MG132 treatment dramatically prevented tumor cells from NRF2 reduction caused by RSL3 in A549 and H2122 cells (Figure 5A). Immunoprecipitation-Western blot assays revealed that RSL3 was able to promote NRF2 protein ubiquitination and degradation in *KLK* LUAD cells (Figure 5B).

### 3.5. RSL3 Directly Targeting USP11 in Promotion of NRF2 Ubiquination in KLK LUAD Cells

Next, we detected levels of *Rbx1*, *Hrd1*, *Cul1* and *Cul3* mRNA and found that their mRNA transcript level was not significantly changed by RSL3 treatment in *KLK* LUAD cells (Appendix A), indicating that other mechanisms could be involved. As we know, USP11 is the recently identified de-ubiquitnase, which was specifically for NRF2, and a previous study showed that its silence destabilized NRF2 protein but sensitized NSCLC H1299 cells to ferroptosis after induction of the oxidative stress [19]. Since USP11 is an enzyme having an active site with cysteine, we speculated that RSL3 might bind to USP11 to inhibit its activity in cells. We conducted the molecular docking analysis to assess the direct binding and found that the catalytic domain of USP11 (USP11-CD) was modeled and predicted to bind to RSL3 with hydrogen (H) bond and hydrophobic interaction (Figure 6A). Moreover, the Gln402, Phe831 and Tyr833 of USP11-CD were also predicted to be the critical amino acid residues to bind to RSL3 (Figure 6A). To experimentally prove it, we first purified wide type and mutant USP11-1 (Y833E, Q402E) proteins using the affinity chromatography (Figure 6B) and performed the Octet platform analysis to detect their biomolecules interactions using the Bio-layer Interferometry (BLI). We confirmed direct USP11 and RSL3 interaction in vitro with a calculated K_D_ of 340 μM (Figure 6C). However, the mutated USP11-1 (Y833E, Q402E) did not lose the ability to bind to RSL3, indicating that Phe831 might be the essential residue for the binding (Figure 6C).

After that, we assessed the inhibitory effect of RSL3 on USP11 activity by establishing that stable USP11 overexpressed A549 and H2122 cells and found that USP11 overexpression was able to stabilize NRF2 protein and maintain NRF2 level before and after RSL3 treatment of *KLK* LUAD cells (Appendix A). Moreover, USP11 overexpression also significantly alleviated NRF2 downregulation induced by RSL3 treatment in *KLK* LUAD cells (Appendix A). USP11 overexpression could significantly relieve NRF2 ubiquitination and degradation induced by RSL3 treatment of *KLK* LUAD cells (Figure 6D), suggesting that RSL3 inhibition of NRF2 expression might be through USP11 suppression in *KLK* LUAD cells.

## 4. Discussion

The effective elimination of cancer cells while leaving the healthy cells intact is ideal and desired state in targeted cancer therapy. Experimental drugs developed based on the synthetic lethality is the strategy to maximally protect normal cells when promoting cancer cell death [22]. A previous study demonstrated that effective inhibition of the SLC7A11-GSH axis was able to induce the synthetic lethality of *Kras*-mutant LUAD, which might be a promising therapeutic approach in control of *Kras*-mutant LUAD clinically [23]. As we know, NRF2 protein was aberrantly highly expressed in *KLK* LUAD cells, speculating as a druggable interaction with *Kras* to generate the synthetic lethality in LUAD cells [5,24,25,26]. Previous studies identified different NRF2 inhibitors by directly or indirectly targeting NRF2 protein and/or the downstream proteins [24,25,26]; however, further assessment of the specific NRF2 pathway inhibition in tumor cells could help us to identify or develop a useful therapeutic approach to *KLK* LUAD. In the current study, we demonstrated the inhibitory effects of the FIN II compound RSL3 on the NRF2-GSH axis in RSL3-induced ferroptosis of LUAD cells in vitro and in vivo. We found that the knockdown of NRF2 expression induced *KLK* LUAD cell sensitivity to RSL3-induced ferroptosis, while RSL3 treatment also reduced expression of NRF2 protein in *KLK* LUAD cells. Moreover, RSL3 treatment suppressed activity of the NRF2-GSH axis during *KLK* LUAD cell ferroptosis occurred in vitro and in vivo. At gene level, RSL3-reduced NRF2 expression was through promotion of NRF2 protein ubiquination in *KLK* LUAD cells. RSL3 was able to directly bind to USP11 enzyme and to promote NRF2 protein ubiquination and degradation in *KLK* LUAD cells. Our future study will assess the RSL3 effects clinically in LUAD patients.

Yang et al. reported that RSL3 targeted GPX4 to induce ferroptosis [14]. GPX4 is the only mammalian enzyme by catalyzing phospholipid hydroperoxides into phospholipid alcohols [27]. To our surprise, our current data showed that the FIN II compound RSL3 was able to inhibit NRF2 expression, whereas a FIN I compound Erastin was unable to change NRF2 level in *KLK* LUAD cells. Molecularly, the RSL3 structure possesses the chloroacetamide moiety, which may be critical for the activity and incline to target activity of the downstream enzymes with a nucleophilic active site [14]. In this regard, the cysteine protease DUBs might be a potential RSL3 target, since their active sites possess a nucleophilic cysteine [20,28]. Indeed, USP11 is a recently identified cysteine protease DUBs as the NRF2 target in NSCLC cells and knockdown of USP11 expression sensitized NSCLC cells to ROS-induced ferroptosis [19]. In the current study, we also identified USP11 as a RSL3 target, i.e., RSL3 molecule was able to directly bind to USP11 protein and in turn inhibited USP11 activity in *KLK* LUAD cells. However, USP11 overexpression was able to prevent the RSL3-induced NRF2 protein ubiquination and degradation, indicating the RSL3 antitumor activity in LUAD by inhibition of NRF2 expression through interaction with USP11 protein. These data suggest that RSL3 possesses a dual role in ferroptosis induction, i.e., on one hand, RSL3 exerts classical ferroptosis by inhibition of the GPX4 activity, whereas in another hand, it can also inhibit the USP11 activity and promote NRF2 proteosomal degradation in *KLK* LUAD cells. Indeed, the GSH synthesis was reduced in LUAD cells in vitro. Withaferin A (WA), a natural anticancer agent, has been shown a high efficacy in inducing ferroptosis by targeting GPX4 but activating HMOX1 to enhance Fe^2+^ concentration [29]. Consistent with their data [29], our current study showed a novel mechanism of RSL3-induced *KLK* LUAD cell ferroptosis in addition to GPX4 inhibition by the dose-dependent induction of *KLK* LUAD cell ferroptosis. The low RSL3 dose could not reduce NRF2 expression, whereas a high RSL3 dose (8 μM) significantly reduced NRF2 expression in *KLK* LUAD cells and this finding let us speculate that a high amount of RSL3 molecules might compete with NRF2 protein to conjugated ubiquitins to bind to USP11 protein, leading accumulation of ubiquitinated NRF2 and entering into the 26S proteasome for degradation. This speculation should be confirmed by the crystal structure analysis of the RSL3-UPS11-NRF2 complex in future study. In the current study, we were able to utilize the Schrodinger software to predict three essential amino acid residues, including Q402, F831 and Y833, that facilitated USP11 protein to bind to RSL3 molecule, whereas mutated Q402E and Y833E did not abolish such an interaction with RSL3 in vitro, but might even promoted their binding. The data may indicate that F831 might be the most important site for USP11 binding to RSL3.

Inducing ferroptosis is considered a novel strategy for the future control of human cancers [30]. Our current study of RSL3 sounds useful, but at this stage, RSL3 is a very toxic compound with a limited level of solubility in water, making it difficult to directly apply with *KLK* LUAD patients in the clinic. To do so, more investigation of RSL3 is needed; for example, Li et al. fabricated Malt-PEG-Abz@RSL3 nanomicelles to achieve more efficient drug delivery to hepatocellular carcinoma cells for ferroptosis [31]. Furthermore, the ubiquitin–proteasome system (UPS) plays an important role in ferroptosis induction [32] and the OTU deubiquitinase (ubiquitin aldehyde binding 1 (OTUB1)) improved SLC7A11 protein stability in ferroptosis [33]. The broad-spectrum DUB inhibitor palladium pyrithione was also able to induce ferroptosis by promoting GPX4 protein degradation in lung cancer cells [34], while erastin promoted the UPS-dependent degradation of VDAC2 and VDAC3 during ferroptosis in melanoma cells [35]. DUBs have increasingly emerged as druggable targets in cancer therapy, but further study of them could help us to identify truly effective ones for our patients. In the current study, we revealed a novel mechanism of FINII RSL3 in the induction of *KLK* LUAD cell ferroptosis through the suppression of the USP11-NRF2-GSH signaling (Figure 7), although the precise action of RSL3 in *KLK* LUAD is needed to further such investigation. In conclusion, this study is the first to have identified USP11 as a target of RSL3, in parallel to GPX4, to induce *KLK* LUAD cell ferroptosis in vitro and in vivo.

## 5. Conclusions

In summary, the study herein revealed a novel mechanism of RSL3 induction in *KLK* LUAD cell ferroptosis by the suppression of the USP11-NRF2-GSH signaling. We for the first time have identified USP11 as a target of RSL3, in parallel to GPX4, to induce *KLK* LUAD cell ferroptosis in vitro and in vivo. Future study will confirm RSL3 as a novel therapeutic approach in the control of *KLK* lung adenocarcinoma.

## Figures and Tables

**Figure 1 cancers-14-05233-f001:**
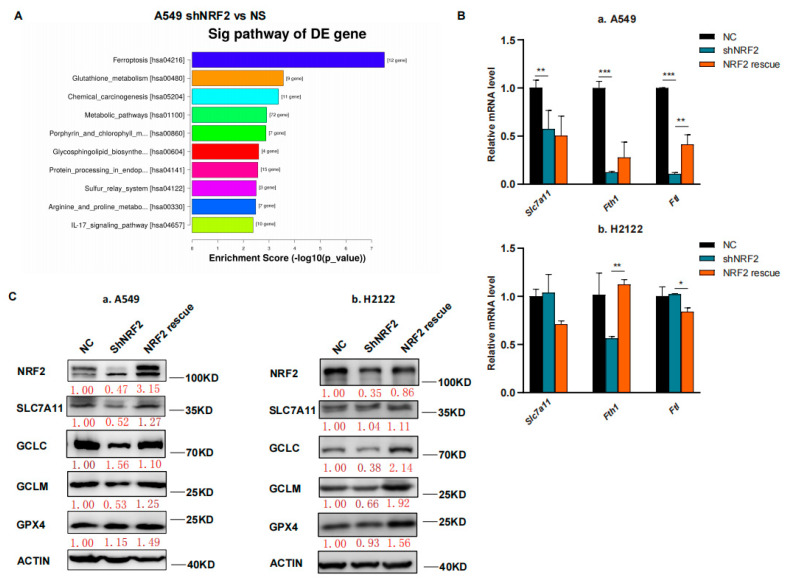
The inhibition of NRF2-GSH axis sensitizes RSL3-induced *KLK* LUAD cell ferroptosis. (**A**) The KEGG analysis. The data showed that the ferroptosis pathway is the top hit in the KEGG pathway enrichment analysis of *KLK* LUAD A549 cells after knockdown of NRF2 expression. (**B**) qRT-PCR. The gene expression involved in ferroptosis was validated using qRT-PCR in indicated cells. *Slc7a11,* solute carrier family 7 member 11; *Fth*, ferritin heavy chain 1; *Ftl*, ferritin light chain. (**C**) Western blot. Protein expression was confirmed using Western blot analysis of NRF2, SLC7A11, GCLC, GCLM and GPX4 in indicated cells. GCLC, Glutamate-Cysteine Ligase Catalytic Subunit, glutamate-cysteine ligase, modifier subunit; GCLM, Glutamate-Cysteine Ligase, Modifier Subunit; GPX4, glutathione peroxidase 4. (**D**) GSH and GSSG assay. The level of changes in GSH+GSSG, GSH and GSSG in indicated cells was assayed. (**E**,**F**) Cell viability assays. A549 cells were treated with dimethyl sulfoxide (DMSO) or RSL3 (4 μM) for 12 h and H2122 cells were treated with DMSO or RSL3 (8 μM) for 12 h and then subjected to CCK-8 assay. Indicated cells were pretreated with DFO (100 μM) or NAC (5 mM) for 4 h (**F**). (**G**) Lipid peroxidation and flow cytometric FACS assays. A549 cells were treated with dimethyl sulfoxide (DMSO) or RSL3 (4 μM) for 12 h and H2122 cells were treated with DMSO or RSL3 (8 μM) for 12 h and then subjected to the FACS assay. * *p* < 0.05, ** *p* < 0.01, *** *p* < 0.001 and **** *p* < 0.0001 versus the control group.

**Figure 2 cancers-14-05233-f002:**
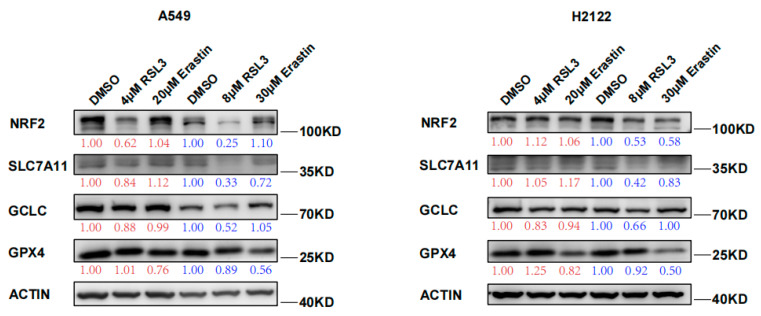
RSL3 suppression of NRF2 protein expression in *KLK* LUAD cells during ferroptosis. Indicated cells were treated with dimethyl sulfoxide (DMSO), RSL3 (4 or 8 μM), or Erastin (20 or 30 μM) individually for 24 h and then subjected to Western blotting analysis of NRF2, SLC7A11, GCLC, GCLM, and GPX4 proteins. Expression change of each protein was quantified by using the Image-Quant LAS 4000.

**Figure 3 cancers-14-05233-f003:**
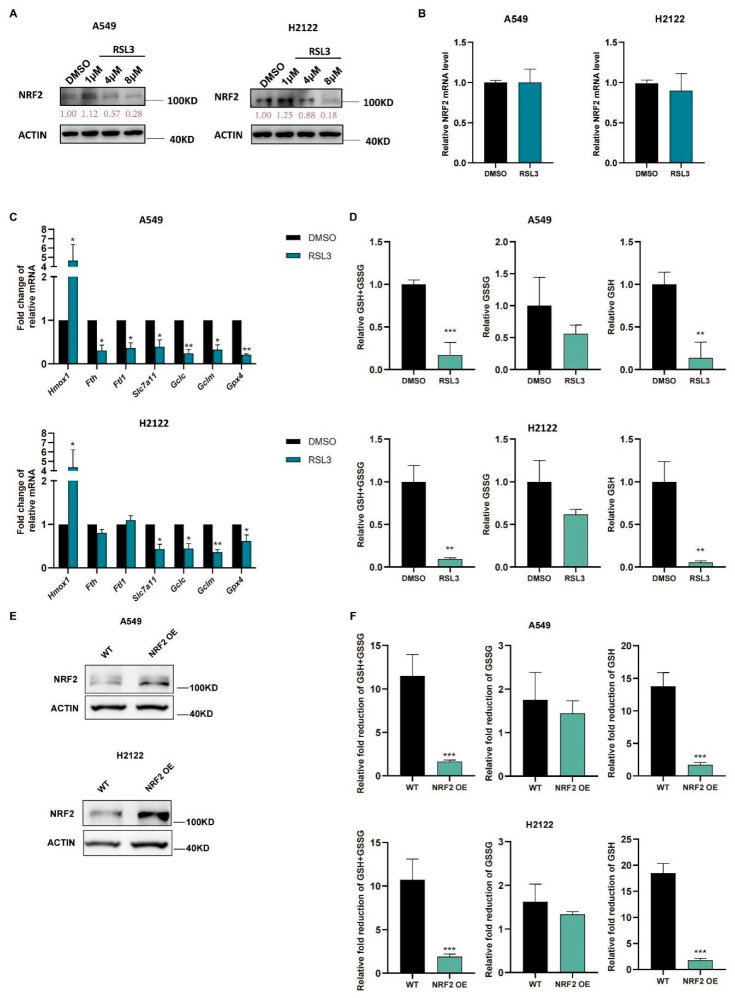
RSL3 inhibition of the NRF2-GSH axis activity during ferroptosis of *KLK* LUAD cells in vitro. (**A**) Western blot. Indicated cells were treated with dimethyl sulfoxide (DMSO)or RSL3 (1, 4 or 8 μM) individually for 12 h and then subjected to Western blotting analysis of NRF2 protein. (**B**–**D**) Indicated cells were treated with dimethyl sulfoxide (DMSO) or RSL3 (8 μM) for 12 h. After treatment, cells were subjected to qRT-PCR analysis of *Nrf2*. (**B**), *Hmox1* (*heme oxygenase 1*), *Fth1*, *Ftl*, *Slc7a11*, *Gclc*, *Gclm* and *Gpx4*. (**C**) or GSH and GSSG analysis of changes in cellular GSH+GSSG, GSH and GSSG levels (**D**). (**E**) NRF2 overexpression was assayed by using Western blot. (**F**) GSH and GSSG assays. The relative fold reduction in GSH+GSSG, GSH and GSSG in indicated cells was assayed. The data are expressed as mean ± SD. * *p* < 0.05, ** *p* < 0.01, and *** *p* < 0.001 versus the control group.

**Figure 4 cancers-14-05233-f004:**
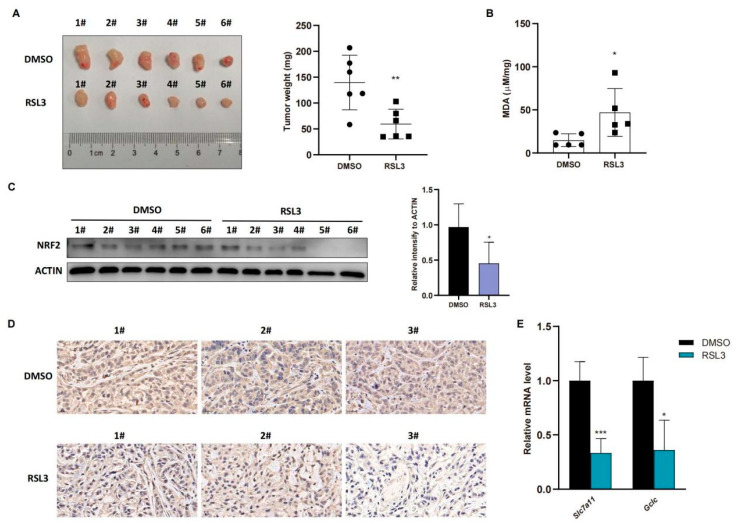
RSL3 inhibition of the NRF2-GSH axis during ferroptosis of *KLK* LUAD cells in vivo. (**A**) Nude mouse tumor cell xenograft assay. Eight-week-old immunodeficient nude mice (n = 6 per group) were subcutaneously injected with cells 5 × 10^6^ A549 cells and treated with RSL3 (100 mg/kg intratumorally, twice a day for one week) after the tumor cell xenograft volume reached 100 mm^3^. After that, mice were sacrificed and tumor weight was measured. (**B**) The malondialdehyde (MDA) analysis. The relative MDA level in tumor cell xenografts was analyzed in indicated group. (**C**) Western blot. Level of NRF2 protein was assayed using Western blot in tumor cell xenografts. (**D**) Immunohistochemistry. Level of NRF2 in tumor cell xenografts was assayed using immunohistochemistry. Magnification factor, 40. (**E**) qRT-PCR. Level of *Slc7a11* and *Gclc* mRNA in tumor cell xenografts was assayed using qRT-PCR. The data were expressed as mean ± SD. * *p* < 0.05, ** *p* < 0.01, and *** *p* < 0.001 versus the control group.

**Figure 5 cancers-14-05233-f005:**
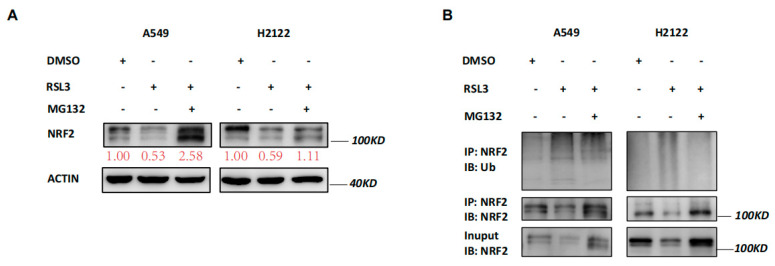
RSL3 reduction of NRF2 expression by promoting of NRF2 protein ubiquination in *KLK* LUAD cells. (**A**) Western blot. Indicated cells were treated with dimethyl sulfoxide (DMSO) or RSL3 (8 μM) for 8 h and then with DMSO or MG132 (25 mM) for additional 4 h and then subjected to Western blotting analysis of NRF2 protein. Expression change of each protein was quantified by using the Image−Quant LAS 4000. (**B**) Immunoprecipitation−Western blot. Interaction of endogenous NRF2 with ubiquitin was assayed, i.e., indicated cells were treated with DMSO or RSL3 (8 μM) for 8 h and then with DMSO or MG132 (25 mM) for additional 4 h and subjected to whole−cell lysis, immunoprecipitation (IP) with an anti-NRF2 antibody, and then Western blotting (WB) with an anti−Ubiquitin antibody.

**Figure 6 cancers-14-05233-f006:**
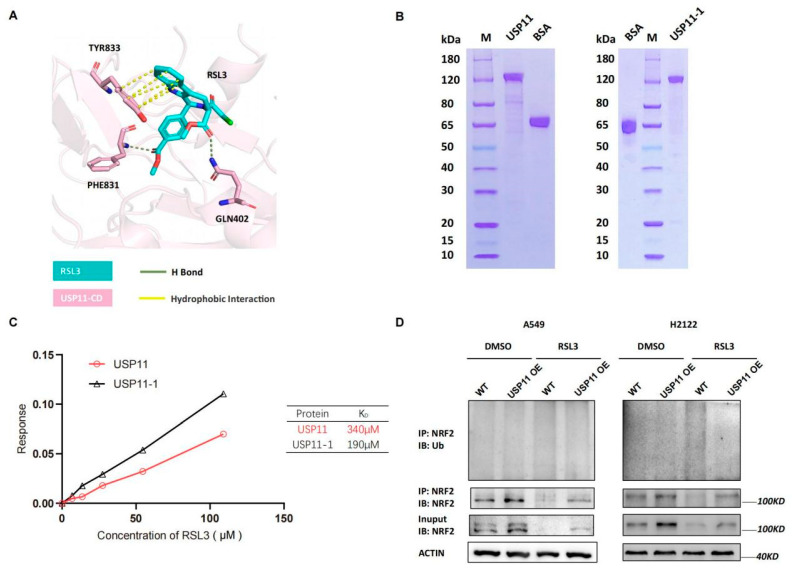
RSL3 directly targeting of NRF2 de-ubiquitinating enzyme USP11 in promotion of NRF2 ubiquination in *KLK* LUAD cells. (**A**) Molecular docking analysis using Schrodinger software. The data showed the interaction of USP11 with RSL3 and the residue of Tyr833, Phe831 and Gln402 was predicted as the essential amino acid residues for RSL3 binding. (**B**) Eukaryotic purification of wild type and mutant USP11-1 (Y833E, Q402E) proteins. (**C**) Affinity assay. We constructed the coding sequences of the wild type and mutant *USP11* (Y833E, Q402E) using PCR and subcloned into the expression vector P35. The resulted vectors were transformed into human embryonic kidney 293 (HEK293) cells for expression of USP11 and USP11-1 proteins, which were then purified by the affinity chromatography. The Octet platform was used to detect and analyze molecule interactions of wild type and mutant USP11-1 with RSL3, based on bio-layer interferometry (BLI). (**D**) Interaction between endogenous NRF2 and ubiquitin under a basal condition and RSL3 treatment in negative control (NC) and USP11 overexpressed *KLK* LAUD cells after treated with dimethyl sulfoxide (DMSO) or RSL3 (8 μM) for 12 h.

**Figure 7 cancers-14-05233-f007:**
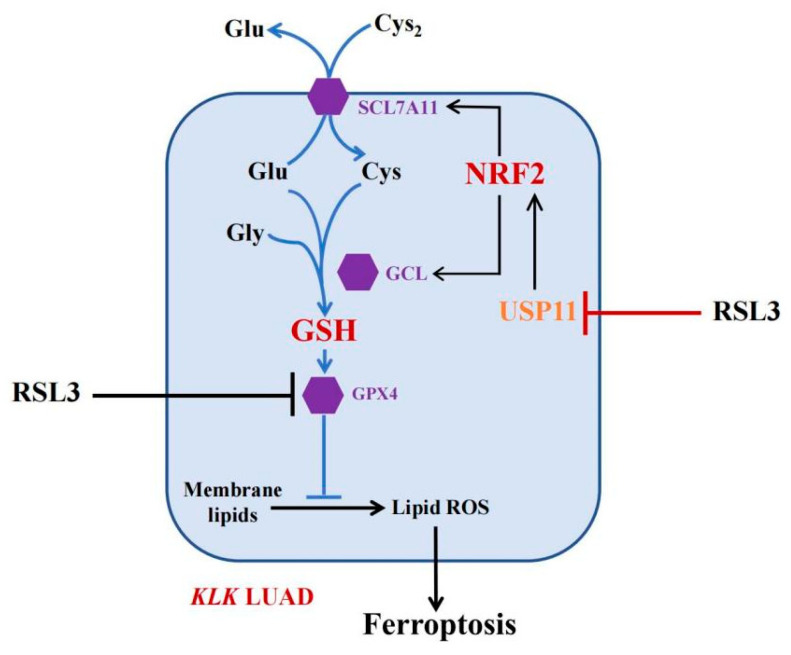
The working model. In *KLK* LUAD cells, RSL3 is able to inhibit multiple gene activities, i.e., in addition to inhibit GPX4 activity, RSL3 can also inhibit USP11 activity, thus induce NRF2 protein ubiquitination and degradation to reduce the GSH production but induce *KLK* LUAD cell ferroptosis.

**Table 1 cancers-14-05233-t001:** Antibodies and main chemicals and reagents used in this study.

Antibodies, Chemicals and Reagents	SOURCE	Identifier
Anti-NRF2 antibody	Abcam	ab62352
Anti-GCLC antibody	Abcam	ab190685
Anti-GCLM antibody	Abcam	ab126704
Anti-GPX4 antibody	Abcam	ab125066
Anti-SLC7A11 antibody	Cell Signaling Technology (Danvers, MA, USA)	12691S
Anti-USP11 antibody	Abcam (Cambridge, UK)	ab109232
Anti-Ubiquitin antibody	Cell Signaling Technology	3936S
Anti-β-actin antibody	Abcam	ab8226
RSL3	SELLECK (Houston, TX, USA)	S8155
Erastin	SELLECK	S7242
DMSO	Sigma-Aldrich	D4540
TRIzol	TIANGEN (Kusatsu City, Japan)	DP424
Fast King gDNA Dispelling RT SuperMix kit	TIANGEN	KR118
TB Green Premix	TAKARA	RR420Q
ECL substrate	Tanon (Shanghai, China)	180–506
CCK-8	Dojindo (Rockville, MD, USA)	CK04
GSH and GSSG Assay Kit	Beyotime (Shanghai, China)	S0053
BODIPY™ 665/676	Invitrogen	B3932
FerroOrange	Dojindo	F374
Corn oil	Abcam	S6701
MDA Assay Kit	Beyotime	S0131S

## Data Availability

The data used in this study are available from the corresponding authors.

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
