# Peer review of "The RSL3 Induction of KLK Lung Adenocarcinoma Cell Ferroptosis by Inhibition of USP11 Activity and the NRF2-GSH Axis"

_cancers, 2022, doi:10.3390/cancers14215233_

Round 1

Reviewer 1 Report

Abstract: Nrf-2 ubiquitination- check spelling

Reduced glutathione (GSH) and oxidized glutathione (GSSG)- abbreviation not described

Fig 1 C-G needs numbering. Fig.1C: Protein expression of NC-Nrf2 vs shNRF2- vs Nrf2 rescue, needs quantification Fig.1G- not legible

The figures showcasing the results have not been explained in details

In vivo Immunohistochemistry results have not been illustrated  

Author Response

Response to Review 1

  1. Abstract: Nrf-2 ubiquitination- check spelling

Response: We did the correction as suggested. Thank you for your kind suggestion.

  1. Reduced glutathione (GSH) and oxidized glutathione (GSSG)- abbreviation not described

Response: We thank the reviewer for pointing out this issue. GSH is the abbreviation of glutathione (r-glutamyl cysteingl +glycine), and GSSG is the glutathione oxidized. We added the description in the revised manuscript.

  1. Fig 1 C-G needs numbering. Fig.1C: Protein expression of NC-Nrf2 vs shNRF2- vs Nrf2 rescue, needs quantification Fig.1G- not legible

Response: We did the numbering as requested. We fully understand the reviewer’s concern and quantified the WB experiment in Figure 1C. To make Fig.1G more legible, we presented the result in a column.

  1. The figures showcasing the results have not been explained in details

Response: We added the explanation in details in figure legends.

  1. In vivo Immunohistochemistry results have not been illustrated

Response: We added the illustration of immunohistochemistry results in the revised manuscript.

Reviewer 2 Report

The authors (Zhang et. al.) of this article entitled ‘RLS3 Induction of KLK Lung Adenocarcinoma Cell Feroptosis by Inhibition of USP11 Activity and the NRF2-GSH Axis’ provide interesting evidence about the mechanisms underlying RLS3-induced ferroptosis in lung cancer cells. Specifically, the authors suggest that RLS3 binds and inactivates USP11 which in turn allows for greater NRF2 ubiquitination and subsequent degradation, leading to altered GSH levels and ferroptosis. Some minor issues should be addressed by the authors to improve the overall quality of the study:

1.    In section 3.1 (page no. 9 first paragraph) the authors write ‘indicating that RSL3 induced ferroptosis resistance of KLK LUAD cells.’. The results of this study do not support such conclusion. Therefore, either more experiments are needed either this should be removed from the text.

2.    In Fig. 3F, the y axis differs from y axis of the respective figures at Fig. 3D. It is recommended to keep the same format to not confuse the reader.

3.    In section 3.3 (page no. 10) the authors write that ‘NRF2 overexpression could significantly alleviate the GSH reduction caused by RLS3 treatment (Figure 3F).’ although Fig. 3F shows that NRF2 overexpression reduces GSH/GSSG. This issue needs to be clarified to avoid confusing the reader.

4.    In section 3.5 Fig. S2 probably refers to Fig. S4 and similarly Fig. S3 refers to Fig. S5.

Author Response

Response to Review 2

The authors (Zhang et. al.) of this article entitled ‘RLS3 Induction of KLK Lung Adenocarcinoma Cell Feroptosis by Inhibition of USP11 Activity and the NRF2-GSH Axis’ provide interesting evidence about the mechanisms underlying RLS3-induced ferroptosis in lung cancer cells. Specifically, the authors suggest that RLS3 binds and inactivates USP11 which in turn allows for greater NRF2 ubiquitination and subsequent degradation, leading to altered GSH levels and ferroptosis. Some minor issues should be addressed by the authors to improve the overall quality of the study:

  1. In section 3.1 (page no. 9 first paragraph) the authors write ‘indicating that RSL3 induced ferroptosis resistance of KLK LUAD cells.’. The results of this study do not support such conclusion. Therefore, either more experiments are needed either this should be removed from the text.

Response: We thank the reviewer for pointing out this issue. We removed the description from the text. We agree that more experiments are needed to do in future to support the conclusion.

  1. In Fig. 3F, the y axis differs from y axis of the respective figures at Fig. 3D. It is recommended to keep the same format to not confuse the reader.

Response: We fully understand the reviewer’s concern. However, Fig.3F refers to the changes brought about by the NRF2 overexpression on GSH and GSSG level. When NRF2 is overexpressed in KLK LUAD cell, the decrease of GSH resulted from RSL3 treatments is alleviated. So the y axis in Fig.3F is different from the y axis of the respective figures at Fig.3D.

  1. In section 3.3 (page no. 10) the authors write that ‘NRF2 overexpression could significantly alleviate the GSH reduction caused by RLS3 treatment (Figure 3F).’ although Fig. 3F shows that NRF2 overexpression reduces GSH/GSSG. This issue needs to be clarified to avoid confusing the reader.

Response: We thank the reviewer for kindly pointing out this issue. We fully understand the review’s concern. As shown in Fig.3F, changes of GSH level was alleviated by NRF2 overexpression upton RLS3 treatment, whereas GSSG level was not. So GSH/GSSG ratio was also changed. We clarified the issue in the revised version.

  1. In section 3.5 Fig. S2 probably refers to Fig. S4 and similarly Fig. S3 refers to Fig. S5.

Response: We thank the reviewer for the carefulness and fixed it accordingly in the revised manuscript.
